# Adolescents’ Perceptions of Sexuality: A Qualitative Study

**DOI:** 10.3390/healthcare11202757

**Published:** 2023-10-18

**Authors:** María Isabel Ventura-Miranda, Andrea Alcaraz-Córdoba, Tania Alcaraz-Córdoba, Guadalupe Molina-Torres, Isabel María Fernandez-Medina, María Dolores Ruíz-Fernández

**Affiliations:** 1Department of Nursing, Physiotherapy and Medicine, Faculty of Health Sciences, University of Almería, 04120 Almería, Spain; mvm737@ual.es (M.I.V.-M.); tac943@ual.es (T.A.-C.); guada.lupe@ual.es (G.M.-T.); isabel_medina@ual.es (I.M.F.-M.); mrf757@ual.es (M.D.R.-F.); 2Distrito Sanitario Almería, Servicio Andaluz de Salud, 04009 Almería, Spain; 3Facultad de Ciencias de la Salud, Universidad Autónoma de Chile, Providencia 7500912, Chile

**Keywords:** sexually transmitted infections, prevention, photovoice, students, sexual health, promotion

## Abstract

Adolescents have a greater risk of acquiring sexually transmitted infections (STIs), which is a serious public health problem. Education is an effective strategy to improve adolescent sexual health outcomes. However, Spanish adolescents have a lack of sex education. The aim of this study was to explore the perceptions and opinions of a sample of adolescents regarding sexually transmitted infections (STIs). Photovoice is a research method that uses the search for images of everyday events with the intention of bringing about social change. An exploratory, descriptive design was used with a qualitative approach based on the Photovoice methodology. The sample consisted of 26 high school and baccalaureate students of Almería (Spain) selected through a convenience sample. The age of the participants ranged from 14 to 17 years, with a mean age of 15.31 years. Two main categories were extracted from the data analysis: ‘Towards a culture of preventing STIs and promoting healthy sexual practices’ and ‘Adolescents’ needs from their perspective’. In conclusion, numerous changes take place at the biopsychosocial level during adolescence that imply a need to explore their sexuality in depth. A lack of knowledge and a carefree attitude during sexual initiation can increase the risk of developing STIs. The study’s adolescents stated that they do not have adequate sex education to acquire sufficient knowledge about sexuality and sexual health, and therefore, request that the traditional format be modified in order to achieve better results.

## 1. Introduction

Numerous studies have identified that adolescents are at increased risk of acquiring sexually transmitted infections (STIs) [1,2,3,4], which is a serious public health problem [5]. STIs are defined as infections that are transmitted from person to person through sexual contact [6] and are preventable through barrier methods of contraception [7] and vaccination [8]. These infections can cause long-term health repercussions, such as infertility, pelvic inflammatory disease, cervical, oropharyngeal or rectal cancer, as well as adverse pregnancy and birth outcomes [9]. Adolescents’ vulnerability to the development of STIs can be associated with social and behavioural factors in particular [10]. Indeed, adolescents are more likely to engage in risky sexual behaviours [11], such as a high number of sexual partners, discontinuous and incorrect use of condoms, exposure to drugs and/or alcohol [12,13] or a need for autonomy and self-affirmation within a peer group [2,14]. In addition, increased risky sexual behaviour has been linked to inadequate knowledge and/or misconceptions regarding STI transmission [2,12,15,16].

Education as a prevention method is an effective strategy to improve adolescent sexual health outcomes [17,18]. However, Spanish adolescents have a low level of sexual education [19]. Sex education refers to the “use of an age-appropriate and culturally relevant approach to teaching about sex and relationships by providing scientifically accurate, realistic and non-judgmental information” [20]. Among the described benefits of using sex education are: increased knowledge and skills about healthy relationships [21], improved self-esteem and self-efficacy for safer sex practices [22], empowerment about one’s reproductive health [23], as well as increased responsible decision-making about one’s sexual behaviour, relationships and reproductive choices [24]. Furthermore, it is the appropriate period to modify acquired beliefs and behaviours, prevent aggressive behaviour, address sexist attitudes and romantic myths and promote gender equality [25,26]. The aim is to ensure that adolescents develop positive sexuality by providing sex education that focuses on preventing and treating STIs, combating sexism and preventing gender-based violence [27].

The Photovoice technique has proven to be a suitable educational method [28] for acquiring theoretical and practical knowledge, as well as improving the understanding of phenomena related to health and care [29]. This technique has been used to explore a variety of community health issues [30], including sexuality education [31]. Photovoice enables adolescents to participate in creating knowledge, expressing perspectives on sexuality in everyday life, formulating perceptions in line with equal rights, as well as rethinking misguided sexual beliefs [32].

Therefore, the aim of this paper was to explore the perceptions and opinions of a sample of adolescents regarding sexually transmitted infections (STIs). This article is important as it contributes to a gap in existing literature.

## 2. Materials and Methods

### 2.1. Study Design

The study design is an exploratory, descriptive design with a qualitative approach based on the Photovoice methodology (“Photo-documentation”, “Photo-elicitation” and “Exhibition in gallery”) and Nola Pender’s theory, which allows us to understand human behaviours related to health, and in turn, encourage healthy behaviours [33]. Photovoice is a research method in which participants are trained to identify, represent and share their realities/experiences through a specific photographic technique [34]. This method allows people to observe everyday events [35] or the reality of others [29,36]. Thus, participants search, select and use photographs to express their perceptions or experiences on a theme. When talking about the selected photographs, participants can explain why they decided to take them and why the images are meaningful or important to them. Photovoice allows researchers to gain a deeper understanding of the topic from the participant’s perspective [37]. It is recommended for use in academic contexts as it stimulates students’ critical thinking; it encourages them to search for photographs, analyse them and discuss what the images represent [38]. The use of this technique can promote social change by, for example, improving community engagement, increasing knowledge of community resources or promoting research self-efficacy [39]. Therefore, this method has allowed us to encourage the participants’ critical thinking and reflection on sexuality, and more specifically, on STIs and methods of STI prevention. In addition, through this technique, students provided relevant information that sparked a dialogue among the group, in which proposals for improvement on the subject were identified. The study followed the Consolidated Criteria for Reporting Qualitative Research (COREQ). The researchers were four women, all trained in sexology. The method of approaching the participants: convenience sampling, due to the availability of the sample [40].

### 2.2. Participants and Data Collection

Data collection was carried out between March and April 2023. The participants were students in the 4th year of Compulsory Secondary Education and the 1st year of Baccalaureate of a secondary school in Almería (academic year 2022–2023). Almeria is located in the south-east of Spain, a province of the autonomous community of Andalusia [41]. The population of Almeria is growing, with a total population of 740.534 [42]. The main economic activity in the Almería region is intensive horticultural cropping [41]. The school belongs to an urban area, has approximately 2500 students enrolled and is a public school. A convenience sample was used to include participants in the study, which resulted in 26 students taking part in the study. The criteria for inclusion were (1) being a student at the school and (2) being willing to participate in the study. The exclusion criterion was refusing to participate in the study. All of the students agreed to participate. The mean age of the participants was 15.31 years (SD: 1.09). The socio-demographic data of the participants are described in Table 1.

In this study, two face-to-face sessions were held. Beforehand, legal guardians were informed by email through the teachers/tutors of each course about the activity to be carried out, its objectives, the voluntary and anonymous nature of participation and the need to provide a consent form signed by legal representatives in order to participate. In the first session, students were informed about the activity to be conducted and the ethical considerations of the study. The need to submit a consent form signed by legal representatives to participate in the second session was emphasised. In addition, explanatory videos on STIs and prevention measures were provided to broaden knowledge on the subject. This session was attended by 26 participants and lasted 45 min.

Two weeks later, the main session was held for three hours, in which a brief theoretical presentation was delivered in order to address any queries they had related to the subject. Next, the participants autonomously selected three photographs through internet searches and social networks, completing a standard form (SHOWED), which is specific to Photovoice. The form asked them to reflect on the following: (i) what you can see; (ii) what is actually happening; (iii) how this relates to our lives; (iv) why the situation in the photo exists; (v) how this image could educate/empower people; and (vi) what we can do about it [43] (Appendix A). Subsequently, five groups of five to six participants were formed, and each member explained the selected photographs and their content. Each group selected a moderator from among its members to facilitate discussion of the photographs and themes. The group members then agreed on a photograph that best represented the current situation regarding STIs and prevention methods. Finally, all participants reflected on the selected photographs and classified them into categories. In this final discussion, the lead researchers acted as moderators. To ensure saturation of themes and categories, the categories were compared and confirmed by all groups of students. The photographs selected in the final phase were displayed on a mural in the students’ classroom for one month. This second face-to-face session was attended by the same participants as the first one. The final discussion was recorded and transcribed verbatim by the principal investigators. In addition, the researchers took notes during the group discussions held prior to the final discussion.

### 2.3. Procedure and Data Analysis

First, a detailed reading of the focus group reflections, the SHOWEDs and researchers’ notes was carried out. Data analysis was performed by two researchers, following a thematic analysis method that contained a series of phases [43]: (1) Familiarisation with the data: the transcripts were read by the researchers to understand everything the participants said. (2) Systematic data coding: the most significant quotes were selected and assigned codes using the “in-vivo coding”, “open coding” and “apply codes” functions in ATLAS. ti. 22. (3) Generation of initial themes from the coded data: initial themes were generated by grouping codes that shared patterns of meaning and had a meaningful relationship around a central idea. (4) Theme development and review: all generated themes and the quotations on which they were developed were double-checked for consistency with the codes they included. (5) Detail, designate and delimit themes: the researchers reviewed the final themes, refined them and created the final names for the themes. (6) Report writing in preparing this research report: the most demonstrative citations were selected. Finally, the researchers clarified the report by filtering out the essential fragments and relating them to the literature review and the aims of the study. Three members of the research team (AAC, MIVM, MDRD) participated in the coding process. As the native language of the participants was Spanish, the discussions were conducted in Spanish and all transcripts/notes were subsequently translated into English prior to analysis.

The COREQ checklist was used to check the quality of the results and guide the writing process [40].

### 2.4. Rigor

To ensure scientific rigour when carrying out the study, the following quality criteria [44] were used. (1) Credibility: the data collection process was detailed; data interpretation was supported by member checking amongst the researchers, and the analytical process was revised by two independent reviewers. (2) Transferability: the study’s setting, participants, context and method were described in detail. (3) Dependability: an expert who did not participate in the data collection and analysis examined the interpretation. (4) Confirmability: all of the researchers read the transcripts independently to ensure that they would reach an agreement regarding the emerging units of meaning, themes and subthemes. Furthermore, some of the participants had the opportunity to see the transcripts [45]. The research team triangulated the data analysis.

### 2.5. Ethical Considerations

The study was conducted according to the guidelines of the Declaration of Helsinki [46] and was approved by the Ethics and Research Commission of the Department of Nursing, Physiotherapy and Medicine of the University of Almeria with registration number EFM 251/23 (Appendix B and Appendix C). The participants and their legal guardians were informed by email about the aim of the study by their teachers/tutors. An information sheet and informed consent form were sent by teachers/tutors, which had to be signed by the students’ legal guardians to allow them to participate in the study (Appendix D). The information sheet explained the voluntary nature of participation in the study, the anonymity of the data collected and the possibility to withdraw at any time. The consent form signed by the legal guardians was given to the principal investigators on the day of the main session. The data obtained were kept confidential and processed in accordance with the Organic Law on Personal Data Protection and Guarantee of Digital Rights [47]. 

Confidentiality and anonymity were guaranteed by replacing the participants’ names with codes. In all phases of the study, the current ethical principles of the Declaration of Helsinki were taken into account [46].

## 3. Results

Two main themes were extracted from the data analysis, ‘Towards a culture of preventing STIs and promoting healthy sexual practices’ and ‘Adolescents’ needs from their perspective’, which helped us to understand adolescents’ opinions or perceptions about STIs and how to prevent them, as well as their views on the approach to sexuality in adolescence (Table 2).

### 3.1. Towards a Culture of Preventing STIs and Promoting Healthy Sexual Practices

This category refers to adolescents’ reflections on STI prevention and barrier methods, as well as personal characteristics (self-esteem, partners, love, etc.) that influence the development of sexual practices.

#### 3.1.1. STIs and Traditional Methods of Protection

The adolescents were able to define STIs, as well as identify examples (HIV, syphilis, chlamydia and gonorrhoea) and possible symptoms (discharge, itching or sores). The students recognised that unprotected sex was a risk factor for STIs and linked the use of prevention methods to good health.


*“STIs are very much related to our health, as they are diseases that we can have and affect us for life...”.*
(S2)


*“...it is important to know the different methods of protection available and how they are used in order to have a good sexual relationship”.*
(FG3)

In addition, they reported that unprotected sexual activity is very common among adolescents. They usually associate this fact, on the one hand, with a lack of knowledge about sexually transmitted infections and the short- and long-term consequences for their health and that of their partners. On the other hand, they link it to careless behaviour in sexual practices.


*“...because they did not know the consequences of not using protection methods or they knew about them, but ignored them”.*
(S10)


*“STIs are very common in young people because they do not use contraceptive methods that reduce their transmission”.*
(S2)


*“STIs are an issue that is very present in our lives, yet there is very little understanding of them”.*
(S1)

The participants referred to the male condom as the most effective and widely used STI prevention method. However, they mistakenly claimed that contraceptive pills are a good method of STI prevention. The adolescents mentioned the importance of using prevention methods appropriately and avoiding practices that reduce their effectiveness.


*“Before having sex, make sure that the condom is on properly, check the expiry date and that it is not defective or broken. They should also be stored in a cool place to avoid breakage”.*
(S20)


*“...if you use condoms you can avoid catching diseases, and it is a popular method because of its effectiveness”.*
(S12)

#### 3.1.2. Love and Self-Esteem as Key Factors of Prevention

The participants stated that the most important factor in preventing STIs is self-esteem and self-respect. They consider self-esteem to be directly related to consent and being able to say no to an unsafe relationship in which protection is not used. The insecurities of adolescents can encourage power relationships and social desirability.


*“(Moderator: How can we prevent STIs?) I feel that mainly loving yourself because accepting something (unprotected sex) that you know is going to be bad for your health means that you don’t love yourself as much as you should”.*
(FG5)


*“...feeling insecure and having low self-esteem can make you have unprotected sex even if you don’t want to... because you want the other person to like you and feel like other people”.*
(FG2)

The adolescents consider that a healthy and loving relationship plays a fundamental role in the prevention of STIs. Thus, they identify love for their partner as a protective factor against STIs, as it encourages the use of prevention methods in order not to harm the other person’s health. However, they also consider that, in some relationships, one partner can intentionally manipulate the other in order to have unprotected sex.

*“The message that this photograph can give us is that if a person loves you, they also care for you and look after what is best for both of you”*.(FG1)


*“We can also protect the people we love... we are the ones who decide what things we share with our romantic partner”.*
(S26)


*“...because I think that, sometimes, there are men who manipulate their partners by telling them not to use protective measures, and force them to have sex without a condom”.*
(S24)

Finally, the participants consider that people should be free to love as long as the physical and emotional well-being of both partners is taken into account. In addition, they stated that healthy sexuality requires sufficient emotional maturity, respect and affective responsibility.


*“It is foolish to ignore what they teach us in lectures (on sexuality), in addition to a lack of respect towards your partner”.*
(FG2)


*“...that all people are free to love and care about any person as long as they are always thinking of both partners’ well-being so that they are healthy and don’t encounter any problems”.*
(S12)

### 3.2. Adolescents’ Needs from Their Perspective

Adolescents have the need to talk about both positive and negative aspects of sexuality, both its benefits and its consequences, if you do not have a healthy sexuality.

#### 3.2.1. Sexuality in Adolescence

The participants refer to sexuality as a very important part of their lives. Yet sexuality, STIs and prevention methods are still taboo subjects in society despite their impact on physical, mental and social health.


*“Sexuality is a fundamental aspect of our physical, emotional and social well-being”.*
(S14)


*“...we only talk about this subject amongst ourselves, I think it’s a bit taboo. I don’t see myself talking about it with my parents...”.*
(FG3)

The participants think that the biggest concern for adolescents when having unprotected sex is the risk of pregnancy, rather than STIs. They consider that they are not vulnerable to STIs as they are usually transmitted among people of a different age or who engage in regular sexual practices.


*“I think that we have a low chance of getting an STI... because this happens to older people or people who are doing it every day”.*
(FG5)


*“Methods of protection were created to prevent unwanted pregnancies, and this is what affects us the most, because it would ruin our lives”.*
(FG1)

Lastly, the adolescents reflected on how they usually do not have an available forum to discuss sexuality, and they do not feel comfortable addressing the issue with their families. To this end, they suggested providing sex education from an early age, facilitating a more natural dialogue with one’s family and creating forums that allow for reflection in educational establishments.


*“It is very important to treat this issue naturally and to provide sex education courses, because this will prevent misinformation and avoid contracting diseases”.*
(S4)


*“Sex education lectures have to start as soon as possible to get this information to as many people as possible”.*
(S14)

#### 3.2.2. Sex Education

The participants feel that there is a lack of knowledge surrounding STIs and methods of protection despite the many resources currently available. They also highlighted how certain myths negatively affect the use of STI prevention methods.


*“Adolescents also have STIs, as people get infected because they are not fully informed about it”.*
(S13)


*“There are also myths surrounding STIs and pregnancy that still lead to unprotected sexual practices”.*
(S2)

In addition, they report that educational interventions are ineffective, as they focus too heavily on theory and are sporadic rather than delivered over time. The adolescents suggested that sex education be improved by addressing this subject regularly, incorporating real cases and practices, using new technologies, as well as facilitating access to the different methods of prevention and diagnosis.


*“Lectures don’t work... my friend has an STI and they have been giving her lectures since secondary school, but she doesn’t pay attention. Coming here, giving a boring lecture and leaving is the same as doing nothing”.*
(FG5)


*“...giving condoms out, bringing people who have an STI to talk about their experience... I mean, real cases, giving more lectures, doing more tests and if they don’t have anything, they can do whatever they want”.*
(FG1)

Finally, the adolescents consider that sex education on the consequences of risky sexual practices and STIs should be further developed; understanding the consequences can provoke feelings of insecurity and fear, which encourages the use of protection during sexual relations. They also highlighted the potential for peer education, noting that it can be a more effective way of engaging adolescents.


*“I think they should show images of what happens when you catch a sexually transmitted disease... show the consequences so that people are scared... and so that they are more cautious”.*
(FG4)


*“... we can talk about it amongst ourselves, help spread the word and raise awareness about using protection amongst our friends”.*
(FG3)

## 4. Discussion

The aim of this paper was to explore the perceptions and opinions of a sample of adolescents regarding STIs. The analysis of the data led to the creation of two main themes: ‘Towards a culture of preventing STIs and promoting healthy sexual practices’ and ‘Adolescents’ needs from their perspective’. The participants learned and developed the ability to identify the main STIs, as well as the warning signs to consult health services. Adolescents with more knowledge about sexuality are more likely to have protected sex, thus reducing the risk of STIs and unintended pregnancies [48]. In addition, greater awareness has been linked to adolescents’ ability to seek information and treatment for possible STIs [49]. However, several authors have found that European adolescents had little understanding of STIs, with the exception of HIV [50,51]. This is highlighted by the development of incorrect and negative attitudes regarding sexuality and its different aspects [52].

The adolescent participants reported that unprotected sex is common in their community. Similarly, in the study by Nathan et al. [53], 69% of the young people surveyed had recently had unprotected sex, and 41% said they were willing to repeat the experience in the future. Early sexual initiation has been associated with an increased likelihood of having unprotected sex [52]. The participants believed that adolescents engage in unprotected sex due to a lack of knowledge about STIs and the consequences for their health, as well as careless behaviour. Previous studies have identified that adolescents’ ignorance about STIs is related to a higher prevalence of these pathologies [49,54,55,56]. However, it is known that having sufficient knowledge about STIs does not always guarantee that a person will behave appropriately [57]. Adolescents also engage in unprotected sex due to the age-appropriate willingness to take risks [58] and not thinking about the consequences [52].

The participants identified the male condom as the most effective and widely used STI prevention method. In this regard, numerous studies have identified the male condom as the method of choice for adolescents [5,48,51]. However, students erroneously identified birth control pills as a good method of STI prevention. Similarly, in the study by Cegolon et al. [51], adolescents selected the contraceptive pill as one of the three best options for preventing STIs. This may be because adolescents associate the use of condoms or barrier methods with pregnancy prevention and do not consider STIs [59].

On the other hand, participants considered having high self-esteem or self-respect as one of the best ways to prevent STIs. Similarly, the systematic review by Ahn and Yang [60] and the study by Sánchez-Sansegundo et al. [61] found a negative relationship between self-esteem and condom use in adolescents. This may be justified because people with low self-esteem focus on avoiding rejection rather than on the possible consequences of casual sexual behaviour [62]. In addition, the students considered that one’s romantic partner plays a key role in the use of protective methods. Similarly, the study of Grubb [5] found higher rates of barrier method use among adolescents whose sexual partners preferred to use barrier methods and who were able to openly communicate their desire to use them. However, some authors report that barrier method use decreases when the partner is perceived as safe, i.e., at low risk of transmitting STIs [63]. In this regard, the participants stated that, in some relationships, one partner manipulates the other to avoid using protective methods. In the study by Davis et al. [64], more than one-third of men and one in six women acknowledged having used emotional manipulation to avoid using condoms during sex.

The participants explained how they do not have forums to discuss sexuality and how it is often a taboo subject at home. The authors Carbonell et al. [65] found that barriers to sex education are associated with both families and educational institutions. Yet, sex education is an effective method of protection against STIs, as it improves knowledge and the use of barrier methods, and it raises awareness of the risks and consequences of unprotected sexual practices [5].

### Limitations

The results of this study should be seen in the light of a number of limitations. Firstly, the ethnic origin of the students was not taken into account, so this aspect should be taken into account in future research. Secondly, the study was conducted in an urban high school, so it would be desirable to replicate this research in a rural area. Future research should consider extending the number of participants to rural schools, the influence of other contextual variables not considered in this study and the comparison of results with other students from all over Spain. Finally, as with any qualitative study, desirability bias may have affected the results and is considered a major limitation of the study.

## 5. Conclusions

In conclusion, the adolescent period is a time of numerous biopsychosocial changes that triggers a need to explore their sexuality in depth. The initiation of sexual activity, combined with a lack of knowledge and a careless attitude, can lead to a risk of developing STIs. The study’s adolescents stated that they do not have adequate sex education to enable them to acquire sufficient knowledge about sexuality. Therefore, they demand a change to the traditional format in order to achieve better results. This study highlights the need to implement sex education programmes that are carried out over time and use innovative methodologies that stimulate and motivate students.

## Figures and Tables

**Table 1 healthcare-11-02757-t001:** Socio-demographic data.

Participants	Age	Sex	Relationship Status	Do You Know Anyone Who Has Had an STI?	Sex Education Received
P1-S1/GF1	15	M	S	No	No
P2-S2/GF1	14	M	S	No	No
P3-S3/GF1	16	F	S	Yes	Yes
P4-S4/GF1	16	F	W	No	Yes
P5-S5/GF1	14	M	S	No	No
P6-S6/GF2	16	F	W	No	Yes
P7-S7/GF2	14	F	S	No	No
P8-S8/GF2	15	M	S	No	No
P9-S9/GF2	14	F	S	Yes	No
P10-S10/GF2	17	M	W	No	Yes
P11-S11/GF2	16	F	S	No	Yes
P12-S12/GF3	14	F	S	No	No
P13-S13/GF3	15	F	S	No	No
P14-S14/GF3	17	M	S	No	Yes
P15-S15/GF3	14	F	S	No	No
P16-S16/GF3	15	F	S	No	No
P17-S17/GF4	14	F	S	No	No
P18-S18/GF4	17	M	W	No	Yes
P19-S19/GF4	16	F	W	Yes	Yes
P20-S20/GF4	16	F	S	No	Yes
P21-S21/GF4	15	M	S	No	No
P22-S22/GF5	16	M	S	No	Yes
P23-S23/GF5	14	F	S	Yes	No
P24-S24/GF5	15	F	W	No	No
P25-S25/GF5	17	M	S	No	Yes
P26-S26/GF5	16	F	S	No	Yes

Note: FG: Final group; S: SHOWED; F = female sex; M = male sex; S = single; W = With a partner.

**Table 2 healthcare-11-02757-t002:** Themes, sub-themes and units of meaning.

Category	Subcategory	Codes	Images
Towards a culture of preventing STIs and promoting healthy sexual practices	STIs and traditional methods of protection	STIs, HIV, syphilis, gonorrhoea, chlamydia, condoms, contraceptive pills, unprotected sex	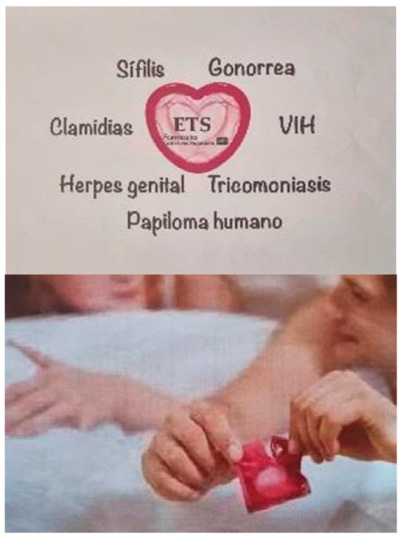
	Love and self-esteem as key factors of prevention	Self-love, loving relationship, maturity, responsibility, respect, danger, protection, prevention, safety, self-confidence, manipulation	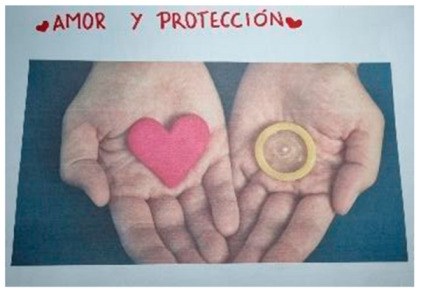
Adolescents’ needs from their perspective	Sexuality in adolescence	Ignorance, taboo, concern, invulnerability, “unnaturalness”, “no spaces”	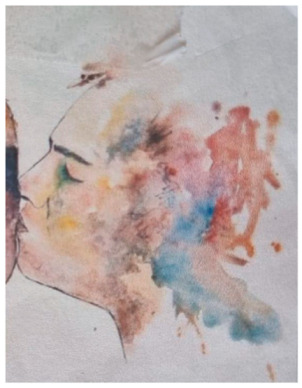
	Sex education	Real cases, lecture, “lectures don’t work”, giving out condoms, diagnosis, new technologies, “giving away means of protection”	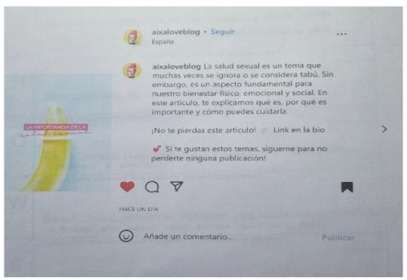

## Data Availability

Data are available from the first author or corresponding author on reasonable request.

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
