# Peer review of "Adolescents’ Perceptions of Sexuality: A Qualitative Study"

_healthcare, 2023, doi:10.3390/healthcare11202757_

Round 1
Reviewer 1 Report
Dear Authors,
Thank you for submitting your manuscript titled "Adolescents' perceptions of sexuality. A qualitative study " to this journal. After reviewing your manuscript, I am pleased to inform you that I have decided to recommend minor revisions before final acceptance.
Overall, the paper presents a valuable contribution to the understanding of adolescents' perceptions of sexuality and STIs, and the use of the Photovoice technique in this context. However, there are some areas that require attention and clarification to enhance the quality of the paper. Below, I have outlined the specific revisions needed:
-
Introduction:
- Clarify the scope of the study more explicitly in the introduction. For instance, specify the research questions or objectives of the study.
- Provide a clear and concise statement of the paper's contribution and significance in addressing the gaps in existing literature.
- Consider adding a brief overview of the structure of the paper, outlining what readers can expect in each section.
-
Methods:
- Clarify the selection criteria for the participants in the study. Explain why the specific age group and school were chosen, and how these factors might influence the findings.
- Provide more detail on how the Photovoice technique was implemented, including any specific prompts or instructions given to participants when selecting and discussing their photographs.
- Describe the process of data analysis in greater detail. How were categories and subcategories derived from the collected data? Were there any challenges or limitations in the data analysis process?
-
Ethical Considerations:
- Mention how informed consent was obtained from participants and their legal guardians in more detail. Specify any ethical considerations related to working with adolescents in this context.
- Provide information on how participant confidentiality and data protection were ensured throughout the study.
-
General Clarity and Formatting:
- Carefully proofread the manuscript for grammatical errors and ensure consistency in formatting and style.
- Check for any incomplete sentences or missing words throughout the manuscript.
-
Thank you for your attention to these matters, and I look forward to receiving your revised manuscript.
Sincerely,
Reviewer
Author Response
Thank you very much for your very pertinent comments and input.

Reviewer 2 Report
Thank you for the opportunity to review this article. I would strongly recommend review of phrasing by a native English speaker. For example, "training on sexuality" (in the abstract) is awkward and somewhat unclear, and "being a serious public health problem" (also in the abstract) is ambiguous in what it is describing (STIs? adolescents' risk?). Please briefly describe/define Photovoice in the abstract. Please also clarify the level of education (and/or age) and the general location of the conveinence sample of Spanish students.
Introduction:
- There are other methods of preventing STIs besides education and barrier devices. HIV PrEP, for example, and vaccination against Hepatitis A and HPV, as others.
- Lines 38-41 are another example of necessary review and editing by a native English speaker. "... such as a high number of sexual partners, inconsistent and/or incorrect use of condoms... or demand for independence and individuality in a group."
- I could be wrong, but I think most English speakers would say "sex education" or maybe "sexual education," not "sexuality education." "Education on sexuality" might work, though.
- Again, Photovoice needs better definition earlier on, though I see an (unclear) explanation in the design section.
Materials and methods:
- The definition of Photovoice is unclear - who is doing the searching? How does that lead to social change?
- Please give more information about Almeria - geographic location, prevailing political attitudes, general age of locals, type of industry/business performed there, etc.
- Table 1 would be much more helpful to me as an appendix, with a summary of the data offered in its place - eg, age range 14-17, 10/26 (38.5%) male, etc.
Results:
- I apologize, but I don't understand how the 2 main themes were defined.
- Table 2 is difficult to understand, as the text is square-justified (rather than left-justified) and the images disrupt reading. I would strongly recommend listing the categories, subcategories, and "units of meaning" (better rephrased as "key words") as the table, with call-outs to the 5 images in their own insert.
- 3.1.1 is missing parentheses when talking about examples of STIs, symptoms, etc.
- The focus seems to be on heterosexual sexual contact - was anything noted regarding male-male or female-male contact and protection?
Discussion:
- Lines 315-316 were already mentioned earlier.
- The limitations section should be marked as 4.1.
- What is meant by "the culture of the students"?
- Desirability bias is a big limitation in interviews on topics such as this.
Conclusions:
- I'm not sure these conclusions state anything new or give a clear direction on next steps in research or education.
Appendices:
- Please provide English translations of all 3 documents. (I can read Spanish but translations should be provided.)
See comments above.
Author Response

(The authors gave the same response as above.)

Reviewer 3 Report
This paper is well-written and offers valuable insights into the unique needs of adolescents. Please see below for suggested edits:
1. The methodology section is well-structured and explains the research design and data collection process. However, including more details about the Photovoice methodology would be helpful. Providing more specifics about this process would enhance the clarity of the methodology section.
2. While knowledge is an important factor in sexual health, knowledge alone may not always lead to behavior change. Therefore, it’s essential to emphasize the need for other evidence-based constructs, e.g., self-efficacy for safer sex practices.
3. Consider including a health promotion theory. It will enhance the paper and have a broader impact on research and practice.
4. Consider positive framing /and inclusive language. For example, consider words other than careless behavior on Ln 305-306.
5. Please consider addressing credibility and transferability. By incorporating these recommendations, you can strengthen the credibility and transferability of your qualitative study, ultimately contributing to its overall validity. I believe these enhancements will further elevate the impact and relevance of your research within the academic community.
6. Consider referring to Consolidated Criteria for reporting qualitative research (COREQ) for further guidance to enhance the manuscript.
Author Response

(The authors gave the same response as above.)

Reviewer 4 Report
Human subjects research on protected populations: The participants in this study are adolescents, all under the age of 18. Children under age 18 are considered a vulnerable group, and special ethical considerations apply. The overwhelming majority of IRBs require parental consent when carrying out research with minors, along with assent from the participants themselves. There is no mention of special protections in place for minors or parental involvement in the consent process. The informed consent form indicates that parents were asked to consent- however, this is in the appendices. The process of collecting informed consent/assent from parents and children should be briefly described in the text of the manuscript.
The appendices are great additions, particularly the SHOWED form and the consent form- these resources are so helpful to researchers doing similar work. I’m very curious about the decision to use the “SHOWED ” English acronym given that the participants were from Spain, and the acronym did not fit the Spanish translation. Were the participants all bilingual? Was the English meaning explained? It would be helpful to have a brief note clarifying this in Appendix 1.
Introduction- Line 33 states “STIs are preventable through health education and barrier methods of contraception”. This is misleading- health education is a strategy to change health behaviors; health behaviors are what improve health outcomes (furthermore, knowledge alone is not enough to change behavior). Please remove health education from this sentence.
Methods:
Section 2.2 Participants and data collection needs more detail
It’s unclear how many meetings were held- typically photovoice studies include at least 2 participant meetings- one to educate participants on the topic, provide basic photography info, and obtain informed consent and then additional meetings to review & discuss images and plan exhibitions. This section should clearly state how many meetings took place, their approximate duration, and how many participants attended each meeting. Photovoice studies typically have a high rate of attrition- often around 50% from the first to final meeting- therefore, demographics from the first meeting are not a reliable indicator of how many participants were part of the discussion.
Line 102 says “groups of 5 or 6 participants were created” – this needs to include how many groups were held and who facilitated these smaller discussion groups. Were discussions recorded and transcribed verbatim or summarized by a notetaker?
Section 2.3 Procedure and data analysis
More detail is needed here also. It’s unclear why the analysis team chose to focus on non-verbal communication. In general, photovoice studies do not consider the images themselves to be data- rather, the photos serve as prompts to generate discussion (which is the data that is analyzed) and tools for advocacy. Therefore, additional details about how the photographs were used in the analysis is important. The process of “extraction of codes and grouped into categories and subcategories” should be described in more detail. How many research team members were a part of the coding process? Were any specific coding & analysis approaches used (citations should be provided if so)? How many rounds of coding were carried out? Was a codebook created? How were disagreements between coders resolved? The authors indicate that “qualitative analysis of the extracted data was carried out….” (line 117). I think this is a typo? Line 113 indicates that the codes are what was extracted, but in most qualitative studies, the data that is analyzed is the text of transcripts or notes.
Results
Table 2 includes “units of meaning” – this is not a term I’m familiar with; I think the authors mean codes?
There are several items in quotation marks – my best guess is that these are in vivo codes taken from the participants’ own words? I’m also confused, because I had the impression that the discussions took place in Spanish? Did the coding process take place in Spanish or were the transcripts/notes translated into English prior to analysis? This should all be discussed in the methods section.
The way quotations from participants are attributed is very confusing. Are the different attributions important for interpreting the results? If so, they should be explained and clarified. If not, a simpler attribution should be used (perhaps simply “participant” or using pseudonyms to distinguish between participants).
Discussion:
Line 292 is very unclear- “Adolescents learned and developed the ability to identify the main STIs as well as the warning signs to consult health services.” – Is this referring to their previous sexuality education experiences? Or to learning that happened directly as a result of viewing the educational materials that were part of the initial Photovoice meeting?
Photovoice is a participatory research method focused on creating change- how were participants engaged in creating change in sexuality education? What are next steps in advocacy/policy change efforts after this study?
It’s hard to understand how the terms “education” and “training” are used throughout the manuscript. It seems “education” refers to sexuality education received by participants; however, I’m uncertain what “training” is –were participants trained to be peer educators or sexuality educators?
The manuscript needs an edit that pays special attention to readability for an English-speaking audience - for example, Table 1 refers to “Sentimental Situation”, when “Relationship Status” is a term that will be much more familiar for most English speaking readers. Similarly, line 98 reads “resolve possible doubts” – “ask questions” is a phrase most English speakers will understand more easily.
Author Response

(The authors gave the same response as above.)
